# A Radiomics-Based Machine Learning Model for Prediction of Tumor Mutational Burden in Lower-Grade Gliomas

**DOI:** 10.3390/cancers14143492

**Published:** 2022-07-18

**Authors:** Luu Ho Thanh Lam, Ngan Thy Chu, Thi-Oanh Tran, Duyen Thi Do, Nguyen Quoc Khanh Le

**Affiliations:** 1International Master/Ph.D. Program in Medicine, College of Medicine, Taipei Medical University, Taipei 110, Taiwan; luuhothanhlam2013@gmail.com; 2Children’s Hospital 2, Ho Chi Minh City 70000, Vietnam; 3City Children’s Hospital, Ho Chi Minh City 70000, Vietnam; thynganchu@gmail.com; 4International Ph.D. Program for Cell Therapy and Regeneration Medicine, College of Medicine, Taipei Medical University, Taipei 110, Taiwan; d151110003@tmu.edu.tw; 5Hematology and Blood Transfusion Center, Bach Mai Hospital, Hanoi 115-19, Vietnam; 6Graduate Institute of Biomedical Informatics, College of Medical Science and Technology, Taipei Medical University, Taipei 106, Taiwan; dtduyen1990@gmail.com; 7Professional Master Program in Artificial Intelligence in Medicine, College of Medicine, Taipei Medical University, Taipei 106, Taiwan; 8Research Center for Artificial Intelligence in Medicine, Taipei Medical University, Taipei 106, Taiwan; 9Translational Imaging Research Center, Taipei Medical University Hospital, Taipei 110, Taiwan; 10Neuroscience Research Center, Taipei Medical University, Taipei 110, Taiwan

**Keywords:** lower-grade glioma, tumor mutational burden, genetic algorithm, radiomics signature

## Abstract

**Simple Summary:**

Lower-grade glioma (LGG) is a kind of center nervous system neoplasm that arises from the glial cells. Lower-grade glioma patients have a median survival time in the range of 1.5–8 years based on the tumor genotypes. In term of epidemiology, most of the lower-grade glioma patients are diagnosed at young adult of age, which led to an early age of death. For exact diagnosis and effective treatment, a pathological result from biopsy sample is required. However, it is long turnaround time. In this study, using pre-operative magnetic resonance images, we developed a non-invasive model to classify tumor mutational burden (TMB), a prognostic factor of treatment response in lower-grade glioma patients, with an accuracy of 0.7936. To our knowledge, our study represents the best model for classification of TMB in LGG patients at present.

**Abstract:**

Glioma is a Center Nervous System (CNS) neoplasm that arises from the glial cells. In a new scheme category of the World Health Organization 2016, lower-grade gliomas (LGGs) are grade II and III gliomas. Following the discovery of suppression of negative immune regulation, immunotherapy is a promising effective treatment method for lower-grade glioma patients. However, the therapy is not effective for all types of LGGs, and tumor mutational burden (TMB) has been shown to be a potential biomarker for the susceptibility and prognosis of immunotherapy in lower-grade glioma patients. Hence, predicting TMB benefits brain cancer patients. In this study, we investigated the correlation between MRI (magnetic resonance imaging)-based radiomic features and TMB in LGG by applying machine learning methods. Six machine learning classifiers were examined on the features extracted from the genetic algorithm. Subsequently, a light gradient boosting machine (LightGBM) succeeded in selecting 11 radiomics signatures for TMB classification. Our LightGBM model resulted in high accuracy of 0.7936, and reached a balance between sensitivity and specificity, achieving 0.76 and 0.8107, respectively. To our knowledge, our study represents the best model for classification of TMB in LGG patients at present.

## 1. Introduction

For decades, the classification of a neoplasm has been based on histological features. These features are inspected through microscopy observation, using hematoxylin and eosin dyes, immuno-histochemical protein expression, and ultrastructural characteristics. Histologically, gliomas are center nervous system (CNS) neoplasms that arise from non-neuronal cells, including the phenotypes astrocytoma, oligoastrocytoma, oligodendroglioma, and glioblastoma [1]. Longing for a narrower differential diagnosis and more accurate classification, the World Health Organization (WHO) revised the 2007 glioma classification in 2016 by including molecular parameters into the scheme [2]. The updated scheme categorizes gliomas into four groups: grade I to grade IV by the level of malignancy in the neoplasm. While in the past, the phrase “low-grade glioma” referred to grade I and II gliomas, a new term in the new scheme, “lower-grade gliomas (LGGs)”, which comprises WHO grades II and III, is gradually becoming more popular. Lower-grade glioma classification plays an essential role in the prognosis of patients [3]. IDH-mutated 1p_19q-codeleted oligodendroglioma is the best prognosis type, with a median survival of 8.0 years. In contrast, the IDH-wild type 1p_19q-non-codeleted subtype is the worst one, with a median survival of only 1.7 years. Notably, some lower-grade gliomas may progress to WHO grade IV glioblastomas within months, whereas a few remain stable for years. Although the WHO 2016 classification of LGGs has been adopted for molecular diagnosis, the known molecular markers are currently limited for explaining the prognosis of LGGs. Thus, further exploration of the genetic mechanism and identification of new biomarkers to predict the prognosis of LGGs is essential for developing precise treatments.

One thing worth noting is the Nobel Prize 2018 was given to James Allison and Tasuku Honjo in the discipline of Physiology or Medicine for their development of the malignancy therapeutic method [4], immunotherapy, which is going to become a widely used cancer treatment. Immune checkpoint blockade such as programmed cell death 1 ligand (PD-L1), indolamine 2,3-dioxygenase (IDO), and cytotoxic T lymphocyte associated antigen 4 (CTLA4), appears to be a potential therapeutic treatment for glioma [5]. However, the biomarker tumor mutational burden (TMB) proved to be a predictive factor of immunotherapy response [6]. TMB, defined as the number of somatic coding mutations per megabase (mutations/Mb) of tumor genotype [7], has become a valuable biomarker across many cancer types for predicting the efficacy of immune checkpoint blockade (ICB) [6,8]. A previous study found that TMB was negatively correlated with overall survival, and a high TMB might inhibit immune infiltration in LGGs [9]. Similarly, Wang et al. [10] concluded that TMB was associated with poor outcomes in diffuse glioma, and a high TMB activated both proliferative activities and the immune responses in gliomas.

Whole-exome sequencing (WES) is considered the gold standard for the measurement of TMB; however, it is not currently used in clinical settings due to high costs and long turnaround times. Developing artificial intelligence models for accurate diagnosis of diseases or biological objects classification, including TMB, to save time and reduce costs is a prominent trend in the modern world. For example, a deep neural network architecture used electroencephalogram (EEG) to discriminate different subtypes of attention deficit hyperactivity disorder (ADHD) [11], a new computer-aided diagnosis system using EEG was built to identify multiple sclerosis (MS) disease [12], and a three-stage deep learning approach was developed to segment red blood cell images and detect malaria [13]. Because TMB is a prognostic factor for immunotherapy, scientists around the world have longed for alternative ways to measure it. In order to classify high and low TMB from lung adenocarcinoma histopathological images, Jain et al. [14] developed a machine learning algorithm, named Image2TMB, that achieved an area under the precision recall curve (auPRC) of 0.92 and an average precision of 0.89, whilst Shi et al. [15] utilized a deep learning model to reach an area under the receiver operating characteristic curve (AUC) of 0.64. Shimada et al. [16] achieved an AUC of 0.91 in developing a convolutional neural network (CNN) model to recognize TMB-High in colorectal cancer patients from hematoxylin and eosin slides.

In our study, we discovered a method representing the best model for TMB classification for LGGs at present. Our study hypothesized that machine learning algorithms could classify high and low TMB using magnetic resonance imaging (MRI) radiomics in LGG patients. Specifically, we proposed a machine learning model, named LightGBM, that can identify TMB groups using preoperative MRIs. To improve model performance, we then utilized the genetic algorithm to pick up the radiomic signature features, which are the most effective for identifying TMB high and low, and fed them into various algorithms to find the best model. Later, we used different imbalanced data solving techniques on our data to reach a good balance between sensitivity and specificity, which gave the most accurate and balanced results at present. Although our study had a small sample size, our results showed that a machine learning model using MRI images can be used to support the clinical situations.

## 2. Results

### 2.1. Patient Characteristics

Table 1 shows the patient’s characteristics of our training and validation cohort. The patients’ data included age, gender, histology, WHO grade, vital status, IDH status, 1p_19q codeletion status, MGMT promoter status, TMB groups (high and low TMB), and subtypes of glioma classification by integration of genetic and epigenetic information, including classic-like, codel, G-CIMP-high, G-CIMP-low, Mesenchymal-like, PA-like. In line with the epidemiology of lower-grade glioma, most of our patients were young adults at the time of diagnosis, with an average age of 44.06 and 49.67 on training and validation dataset, respectively. This difference in age between training and validation data is statistically significant with *p* < 0.05. In addition, most of our patients had IDH mutant (84 mutants versus 21 wildtypes) and 1p_19q non-codeletion status (78 non-codel versus 27 codel) in their genomic information. Next, there were fewer patients with the MGMT promoter in the unmethylated group in the training cohort (12 unmethylated versus 51 methylated) and validation cohort (3 unmethylated versus 39 methylated), and the difference between the two groups in both cohorts was statistically significant with *p* = 0.045106. According to the data characteristics, a higher number of TMB low group compared to TMB high group occurred in training (38 TMB low versus 25 TMB high) and validation data (24 TMB low versus 18 TMB high). By observing the *p*-value, data statistics showed a consistent level between our experiment’s training data and validation data.

### 2.2. Baseline Comparison and Radiomics Signature Building

In the feature selection process, we used different machine learning-incorporated genetic algorithm (GA) models to pick the most effective features, called GA features, from the 726 features in the feature extraction stage. Then, these GA features were evaluated by six different machine learning models (logistics regression—LR, support vector machine—SVM, random forest—RF, linear discriminant analysis—LDA, light gradient boosting machine—LGBM, and extreme gradient boosting—XGB) in predicting the TMB group. The GA features selection and model performance results are given in Table 2.

As shown in Table 2, the LGBM model achieved the best performance with a sensitivity = 0.72, specificity = 0.8893, precision = 0.8367 and accuracy = 0.8218, with 11 GA features. The LGBM model not only reached the highest accuracy among the six GA-based machine learning algorithms in predicting TMB groups, but also showed a balance between sensitivity and specificity. An older version of the LGBM algorithm, the XGBoost algorithm, also showed a high accuracy of 0.7808 with 7 GA features in our experiment. The RF model yielded the second-best performance with an accuracy of 0.8089 with 6 GA features. The LR showed a good result with an accuracy of 0.7936 and its precision of 0.836667 was equal to the best model, LGBM. The worst performance came from the other two models, SVM and LDA, with an accuracy of 0.7462 and 0.7449, respectively; however, this was an acceptable level for a classification challenge. We also performed McNemar statistical tests to see the significant differences between our selected model (LightGBM) and the other models. The results then showed that LightGBM was significantly better than the other algorithms in terms of sensitivity. Moreover, its accuracy was also superior to that of the SVM, LDA and XGBoost models. For two other metrics (specificity and precision), there was not any significant difference among all algorithms. However, these statistical tests showed that LightGBM was the optimal choice for this prediction task with a high performance. Since the LightGBM model was the best model with 11 optimal features, which were chosen by the GA, we considered these 11 GA features the radiomics signature of TMB prediction in our study.

### 2.3. Imbalance Solving

Since the light gradient boosting machine (LGBM or LightGBM) was the best model for predicting the TMB group in our experiment, the imbalanced data solving techniques were then applied to the LightGBM model. The SVMSMOTE technique was the optimal one as shown in Table 3. The SVMSMOTE algorithm showed the highest accuracy (0.7936) and precision (0.7952) among the six different sampling strategies in predicting the TMB group using the LightGBM model. A good balance between sensitivity and specificity, 0.76 and 0.8107, respectively, was recorded by the SVMSMOTE.

### 2.4. Model Interpretation

For understanding the effectiveness of the GA features for the machine learning performance, the shapley additive explanations (SHAP) was conducted to interpret the LightGBM model. In the SHAP analysis, we plotted the value dots of each GA feature via the impact on model output by that feature (known as SHAP value) on the horizontal axis. The rank of the associated feature was determined by the SHAP value. The color-coded vertical axis, which spanned from blue to red, reflected the value of a feature from low to high. It is important to note that the SHAP estimates how important a feature is by seeing how well the model performs with and without that feature for every combination of features. As shown in Figure 1A, features “TEXTURE_GLCM_NET_FLAIR_Dissimilarity” and “HISTO_NET_T2_Bin4” contained many red dots on the positive side of the SHAP value. This indicated that these two GA features at high values have a significant contribution to the model output. Reversely, features “DIST_Vent_TC”, “ECCENTRICITY_NET”, “TEXTURE_GLSZM_NET_T1_SZE”, “TEXTURE_NGTDM_NET_T1_Contrast” and “TEXTURE_GLOBAL_NET_T1_Kurtosis” had most blue dots on the positive side. This meant that these five GA features contributed to the final result at low values. The rest of the GA features “HISTO_NET_T1Gd_Bin10”, “TEXTURE_NGTDM_NET_FLAIR_Contrast”, “TEXTURE_NGTDM_NET_T1Gd_Complexity”, and “TEXTURE_GLSZM_NET_T1_HGZE” appeared with all dots around the zero point of the SHAP value. We considered that features that appeared with color dots around the zero point did not support the model performance significantly. The observation in Figure 1A showed out that TEXTURE_GLCM_NET_FLAIR_Dissimilarity and HISTO_NET_T2_Bin4 were the most productive features in predicting the TBM of the LightGBM model.

Next, we investigated the correlation of a feature amongst two TMB groups. The two first-rank features from the SHAP analysis, TEXTURE_GLCM_NET_FLAIR_Dissimilarity and HISTO_NET_T2_Bin4, were observed for the entire 105 TMB participants. Figure 1B expressed the box plots of our investigations with number 0 standing for TMB low and number 1 standing for TMB high. Since the boxes of TMB high (orange box) and TMB low (blue box) overlapped together in both features, the median lines in the boxes were considered. The median line of TMB high of the feature “TEXTURE_GLCM_NET_FLAIR_Dissimilarity” lay completely outside of the TMB low’s box in the box plot graphic, whilst the second feature, “HISTO_NET_T2_Bin4”, had the median line of the TMB low group located lower than the Q1 (quartile 1) line of the TMB high group. The results in Figure 1B showed that the two first-rank features were significantly different in the TMB high and low group. This led to the effective contribution of two features to our model performance. Obviously, the results by SHAP and bot plots were consistent evidence that the two first-rank features, TEXTURE_GLCM_NET_FLAIR_Dissimilarity and HISTO_NET_T2_Bin4, contribute effectively to our model output.

### 2.5. Validation Results

To ensure the efficiency of our final model, we coded in the Python environment program to draw the ROC and PR curve. This experiment compared training and validation data to evaluate the predictive performance of the LightGBM model and radiomics signature. In detail, we fed the radiomics signature from the training and validation dataset and inserted them into our model. Then we showed the comparative performance of two cohorts in terms of area under the ROC (auROC) and PR curve (auPRC) as in Figure 2. In Figure 2A, the training and validation data resulted in 0.8214 and 0.7857 for auROC, respectively, whilst in Figure 2B, these were 0.7596 and 0.7556 for auPRC. We observed that they were consistent between these two sets, which ensured that our model was reliable and did not contain too much overfitting. It also means that our 11-radiomics signature might be significant in classifying the TMB group.

## 3. Discussion

LGGs, which have a better prognosis than glioblastomas, have a median survival time in the range of 1.5–8 years based on their molecular subtypes [3]. Most of the LGG patients were diagnosed as a young adult, which led to an early age of death. Many studies on mutated genes in LGG patients have improved the effectiveness of treatment for these patients, including surgical resection, chemotherapy and radiotherapy [2,17,18,19,20]. With the development of immunotherapy in discovering the immune checkpoint inhibitors, the classification of LGGs into high and low TMB expanded the therapeutic possibilities and prognosis of the disease.

LGGs are suspected by clinical symptoms such as seizure and headache, and then diagnosed by a combination of medical images, histopathologic results, and molecular characteristics. MRI is an essential imaging method to identify and locate a mass to diagnose lower-grade gliomas and many neoplasms. A number of studies have been published about using MRI to predict LGG models regarding genotypes. Li et al. [21] used a conventional MR-based nomogram model to classify LGGs into three molecular subtypes: IDH mutation and 1p_19q codeletion, IDH mutation and 1p_19q non-codeletion, and IDH wildtype. Yan Ren et al. [22] used the apparent diffusion coefficient (ADC) model to predict IDH1 and ATRX genes in LGGs. Regarding tumor mutational burden, a number of studies were conducted to predict TMB groups amongst different cancers: lung adenocarcinoma [14,15], colorectal cancer [16], and bladder cancer [23]. Notably, we found only one study by Liu et al. [24], whose research recognized high TMB in lower-grade gliomas. Table 4 summarizes the results of the aforementioned studies. To our best knowledge, we are the first to utilize machine learning models to investigate the correlation of MRI features and TMB in LGGs.

Applications of radiomics have attracted a lot of interest recently due to their ability to offer useful interpretive and predictive data for directing treatment options. Additionally, in order to develop more precise prognoses and therapy responses, a variety of artificial intelligence (AI) techniques, including machine learning and deep learning, have been employed to unravel relationships between clinical symptoms and genetic characteristics. In term of machine learning, data exclusiveness is an important process to develop a robust radiomics-based algorithm. In the feature extraction strategy, a huge number of features are retrieved from MRI images, CT scans, gene patterns, protein sequences, etc. Then, these features are fed into the machine learning algorithms. However, varied features provide different contributions to predictive models; some elements strengthen the outputs, while others lessen the predictive model’s potency. Therefore, the results will be greatly influenced by choosing the appropriate features. There are different methods to select the appropriate features. For instance, Le et al. [25] used the Spearman correlation test and F-score analysis to find significant features for glioblastoma identification. Kha et al. [26] employed SHAP analysis, a machine learning model, to find a radiomics signature for 1p/19q codeletion status prediction. A novel method for feature selection was conducted in our study where an incorporated model of the genetic algorithm (GA) and machine learning algorithms was developed to identify the radiomics signature. Specifically, six machine learning algorithms including LR, RF, SVM, LDA, LGBM and XGB were respectively integrated into the GA to find the most effective radiomic features for TMB classification in LGGs, which were then called GA features. We considered the two justifications offered regarding the usefulness of the GA features. Firstly, the genetic algorithm is an evolution system in which the population’s best members are chosen in order to produce the following generation, which could have a more potent solution. Secondly, random mutations and crossover events during the process of evolution produce stronger individuals. The GA–machine learning integrated models were able to choose the most crucial radiomics features based on the fitness function, crossover, and mutation processes by simulating natural phenomena.

In our experiments, the LightGBM model outperformed the others with 11 radiomics signatures (Table 2). In SHAP analysis, to understand the correlation of the GA features and TMB groups, the features TEXTURE_GLCM_NET_FLAIR_Dissimilarity and HISTO_NET_T2_Bin4 appeared as the two first-rank features (Figure 1A). From interpretation of the correlation between these two first rank features with TMB groups in both plots, the disparity between the TMB high and low group of the two key features proved the effective contribution of them to the model performance (Figure 1B). Since radiomic features were extracted from MRI images, biological characteristics of features on images should be considered. In MRI images, LGGs represent with low signal intensity on T1-weighted sequence; reversely, they show high signal intensity on T2-weighted and fluid attenuated inversion recovery (FLAIR) sequence. The intensity of the LGG image is often homogeneous in MRI sequences. In term of radiomics, a gray-level co-occurrence matrix (GLCM) that reflects spatial intensity (gray-level) correlations and distributions of voxels is a second-order statistic of the textural feature set. While a histogram-based feature illustrates the frequency distribution of intensity values that occur in an image. These medical imaging characteristics explained one textural gray-level co-occurrence matrix (GLCM) feature, TEXTURE_GLCM_NET_FLAIR_Dissimilarity, and one histogram-based feature, HISTO_NET_T2_Bin4, appearing as the radiomics signatures for LGG recognition in our algorithm.

There are several limitations of this research that also need to be discussed. Imaging features from only a small number of patients were extracted for the radiomics investigation, which was one of its shortcomings. Overfitting and issues with data dimensionality resulted from it. Moreover, the issue of imbalanced data was also addressed in our study. To deal with these challenges, both undersampling (RandomUnderSampler) and oversampling (ADASYN, BorderlineSMOTE, RandomOversampler, SMOTE, SVMSMOTE) methods were applied on our model. The SVMSMOTE technique, a variant of SMOTE which uses a support vector machine (SVM) algorithm to detect which sample to use for generating new synthetic samples, significantly improved our model performance (Table 3). In comparison to previous studies, although the proposed study showed the best performance on TMB in lower-grade glioma, the prediction of TMB in LGG patients is still not as high as other cancers. We believe that our model can be improved in the future when we harvest more samples to put into the model.

## 4. Materials and Methods

### 4.1. Patient Cohort

The Cancer Image Archive (TCIA) [27], a public online archive with a substantial collection of cancer medical images, served as the patient cohort for this study. Data from TCGA-LGG project patients with LGG were examined. One hundred and ninety nine LGG patients are represented in this collection, together with their MRI scans and clinical data. Patients whose data did not satisfy all of the following criteria were excluded: (1) grade II or III glioma histopathology according to WHO 2016 criteria; (2) sufficient genomic information in TCGA-LGG project recording; (3) pre-operative multimodal magnetic resonance imaging (mMRI) data, including T1-weighted pre-contrast, gadolinium-enhanced T1-weighted (T1-Gd or T1 post-contrast), T2-weighted (T2) and T2 fluid attenuated inversion recovery (T2-FLAIR). Finally, 105 people were included in our study for the future analyses and the construction of machine learning models. The entire dataset was then divided randomly into two groups: 42 individuals for the testing dataset and 63 individuals for the training dataset.

In our study, the tumor mutational burden (TMB) information was collected from a previous study [10], which investigated the TMB in association with the prognosis of glioma patients. The authors analyzed the area under the receiver operating characteristic curve (AUC) for TMB on survival time using the X-tile software [28]. The TMB cut-off value (0.655 mutations/Mb) was the optimal value at which the TMB was significantly associated with poor outcomes in lower-grade glioma, while the glioblastoma was not. This cut-off value of TMB in glioma patients was divided into TMB-High and TMB-Low groups and then applied in our study.

### 4.2. MRI Segmentation and Radiomics Feature Extraction

The authors of a prior study [29] who segmented MRIs into three parts and extracted the associated radiomics features for each region provided the features that were used in this collection. GLISTRboost, a computer-aided segmentation labels program, was used to segment the MRI images of the patients. A hybrid generative-discriminative model makes up the GLISTRboost program. The enhancing part of the tumor core (ET), the non-enhancing part of the tumor core (NET), and peritumoral edema were identified as three sub-regions of a tumor by the expectation-maximization (EM) system in the generating component (ED). The hyperintensity compared to the normal/healthy white matter of T1 and T1-Gd distinguishes the enhanced portion of the tumor core (ET) visible in T1-Gd. On the other hand, a region of the tumor core known as the non-enhanced component of the tumor core (NET) does not exhibit any enhancement, particularly non-enhanced transitional/pre-necrotic or necrotic regions. In contrast to the normal/healthy white matter of T1 and T1-Gd, the NET has a hypo-intensity appearance in T1-Gd. Peritumoral edema (ED), the outside region of the tumor core, exhibits a hyper-intensive signal on FLAIR sequences. To enhance the patient tumor information, the discriminative aspect of the GLISTRboost program was constructed from a gradient boosting multi-class classification algorithm. A manual revision was performed to ensure the corrected segmentation labels by the computer. Figure 3 gives an example of the segmentation technique.

A sophisticated panel to extract radiomic features from segmentation images was also provided by the GLISTRboost software. There are 726 radiomic characteristics in this panel, including the following: (1) intensity; (2) volumetric [30]; (3) morphologic [31]; (4) histogram-based [32]; (5) textural parameters, including features based on wavelets [33], grey level co-occurrence matrix (GLCM) [34], gray-level run-length matrix (GLRLM) [31], gray-level size zone matrix (GLSZM) [35], neighborhood gray-tone difference matrix (NGTDM) [36]; (6) spatial information [37] and parameters extracted from glioma growth models [38]. In order to be published in the TCIA database, these radiomic properties are required to pass the TCGA program’s quality evaluation [29].

### 4.3. Feature Selection and the Genetic Algorithm (GA)

In order to keep features with a high degree of separability between two classes and eliminate noisy variables, feature selection is crucial. This results in predictions that are more accurate. We performed a radiomics feature selection process using different machine learning algorithms that integrated a genetic algorithm (GA) to find out the most effective features for the identification of the TMB group. Each of six machine learning algorithms: logistic regression (LR), random forest (RF), support vector machine (SVM), linear discriminant analysis (LDA), light gradient boosting machine (LGBM), extreme gradient boosting (XGB), were integrated into the GA to discover the most appropriate subset of features that contributed to improving the tumor mutational burden prediction. The dataset with 726 radiomics features was used as the input in this experiment.

The genetic algorithm, a metaheuristic method based on the theory of evolution, can produce effective models when the space of possible solutions to a problem is too enormous to measure. This approach starts with a set of randomly produced solutions, such as “chromosomes” or “genomes”, which are then chosen based on a set of predetermined standards (i.e., fitness function). The greater the degree to which the solution satisfies those requirements (i.e., the fittest parents), the more likely it is to be picked for a crossover with other selected parents. New generations are produced by a process known as “the mating process”. In this study, fitness was defined as the accuracy as determined by the machine learning model integrated in the GA. Newly developed solutions (i.e., the offspring) are more likely to inherit phenotypes that account for an increase in the fitness score. Individual solutions with higher fitness scores may produce “better” offspring through the crossover process with others during the GA evolution process (i.e., generation after generation). The “fittest” solution to the issue might finally emerge after many generations.

### 4.4. Machine Learning

Different machine-learning models were implemented in this research to see which algorithms perform well for these forms of radiomics. These included six machine learning algorithms: LR, RF, SVM, LDA, LGBM and XGB. Our machine learning models were implemented using Python programming language and scikit-learn library [39]. Among the six different classifiers, LR and LDA are simple techniques borrowed by machine learning from the field of statistics. To make predictions in LDA and LR models, we used Bayes’ Theorem and plugging numbers into the logistic regression, respectively. In more complicated algorithms, SVM can identify the most effective hyperplane for discriminating different targets and transform a non-linearly distributed feature space into a high-dimensional feature space. RF, LGBM and XGB algorithms are ensemble learning techniques that collect individual outcome predictions from numerous weak learners and select the final model based on the votes. Hyperparameter tuning was performed on the aforementioned algorithms using grid search on cross-validation. The optimal hyperparameters of all algorithms are shown in Table 5. These six models were incorporated into the genetic algorithm, and the model with the highest performance was chosen for further analysis. The predictive accuracy of the training-testing dataset was evaluated using the five-fold cross-validation method.

The imbalance data problem, which also occurred in our study, is a common difficulty when using machine learning classification. Consequently, this causes a disparity between sensitivity and specificity in the TMB group prediction of our models. Using Python programming language coding, we exploited different techniques for controlling the imbalance challenge: adaptive synthetic (ADASYN) algorithm [40], synthetic minority oversampling technique (SMOTE) [41], borderline-SMOTE [42], SVMSMOTE [43], random over sampler (RandomOverSampler) and random under sampler (RandomUnderSampler) [44].

### 4.5. Statistical Analysis

Student’s *t*-test and the Mann-Whitney U test were performed to compare continuous and categorical variables between the training and validation cohorts. The *p*-value < 0.05 was considered statistically significant. Moreover, our problem is a binary classification. Therefore, each machine learning model’s performance was examined via *sensitivity*, *specificity*, *precision*, and *accuracy*. These evaluation measurements are defined in previous radiomics works [25,45] as follows:(1)$$Sensitivity=\frac{TP}{TP+FN}$$
(2)$$Specificity=\frac{TN}{TN+FP}$$
(3)$$Precision=\frac{TP}{TP+FP}$$
(4)$$Accuracy\;\left(Acc\right)=\frac{TP+TN}{TP+FP+TN+FN}$$
where *TP*, *TN*, *FP* and *FN* denote true positives, true negatives, false positives, and false negatives, respectively. Moreover, to overcome the possibilities of the imbalance dataset, we reported the ROC curve and PR curve to observe the overall model performance. All statistical analyses in our study were conducted using Python programming language coding.

## 5. Conclusions

We investigated MRI-based radiomics features in relationship with tumor mutational burden in lower-grade gliomas; our model is the first to apply the machine learning method. By mimicry of the theory of evolution, the LightGBM-GA incorporation picked up 11 radiomics signatures for TMB classification. Our LightGBM model reached promising results with a high accuracy of 0.7936, and a balance between sensitivity and specificity (0.76 and 0.8107, respectively).

## Figures and Tables

**Figure 1 cancers-14-03492-f001:**
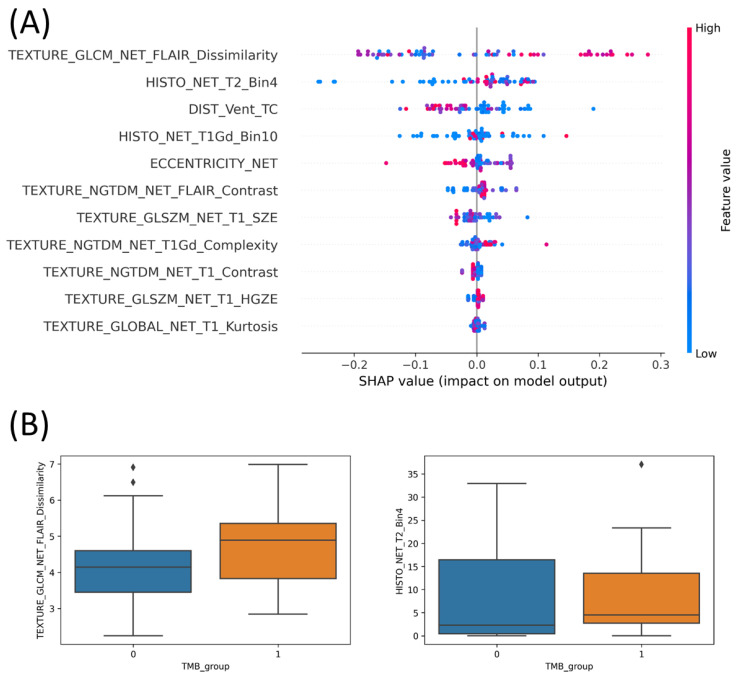
LightGBM model interpretation. (**A**) SHAP analysis on 11 optimal features, (**B**) correlation of two first-rank features with the TMB group. All of these model interpretation experiments were coded in Python programming language environment.

**Figure 2 cancers-14-03492-f002:**
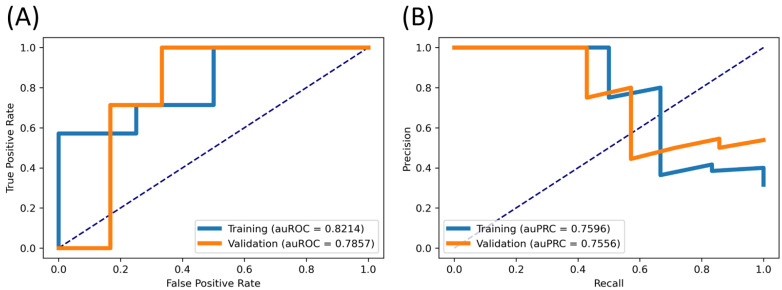
Comparative performance between training and validation data. (**A**) ROC curve, (**B**) PR curve.

**Figure 3 cancers-14-03492-f003:**
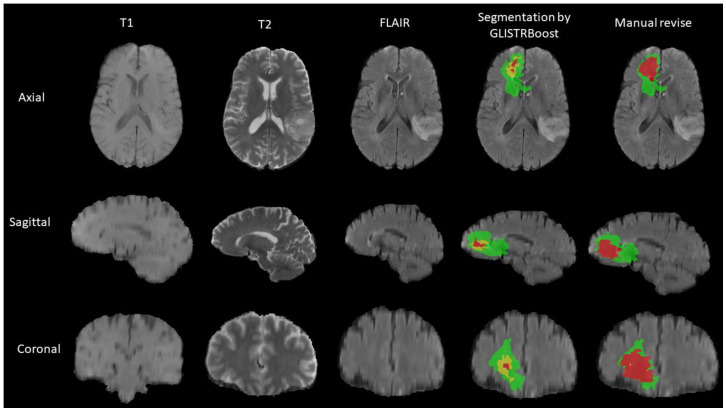
An example of MRI segmentation. The enhancing part of the tumor core (ET) in yellow, the non-enhancing part of the tumor core (NET) in red, and the peritumoral edema (ED) in green.

**Table 1 cancers-14-03492-t001:** Patient characteristics.

		Training	Validation	*p*
Age		44.06 ± 14.00	49.76 ± 13.44	0.040399 *
Gender	Male	26	23	0.088934
	Female	37	19	
Histology	Astrocytoma	21	11	0.127063
	Oligoastrocytoma	18	10	
	Oligodendroglioma	24	21	
Grade	II	26	20	0.262543
	III	37	22	
Subtype	Classic-like	2	3	0.09987
	Codel	13	15	
	G-CIMP-high	38	16	
	G-CIMP-low	2		
	Mesenchymal-like	6	7	
	PA-like	2	1	
Vital status	Dead	13	13	0.117054
	Alive	29	50	
IDH status	Mutant	53	31	0.099583
	Wildtype	10	11	
1p_19q codeletion status	Codel	13	14	0.073916
	Non-codel	50	28	
MGMT promoter status	Methylated	51	39	0.045106 *
	Unmethylated	12	3	
TMB group	TMB high	25	18	0.37498
	TMB low	38	24	

* statistically significant with *p* < 0.05.

**Table 2 cancers-14-03492-t002:** Comparative performance among different GA-based machine learning algorithms in predicting the TMB group.

Algorithm	GA Features	Sensitivity	Specificity	Precision	Accuracy	Running Time (s)
LR	13	0.64 ± 0.265	0.9 ± 0.079	0.8367 ± 0.162	0.7936 ± 0.132	0.159057
SVM	10	0.56 ± 0.15	0.8714 ± 0.177	0.7733 ± 0.228	0.7462 ± 0.133	0.050138
RF	6	0.64 ± 0.15	0.9179 ± 0.064	0.8833 ± 0.108	0.8089 ± 0.041	0.817011
LDA	4	0.56 ± 0.16	0.8714 ± 0.131	0.77 ± 0.131	0.7449 ± 0.056	0.125044
LGBM	11	0.72 ± 204	0.8893 ± 0.131	0.8367 ± 0.131	0.8218 ± 0.1	0.094271
XGB	7	0.6 ± 204	0.9 ± 0.009	0.8 ± 0.106	0.7808 ± 0.08	1.63018

GA: genetic algorithm, logistic regression (LR), random forest (RF), support vector machine (SVM), linear discriminant analysis (LDA), light gradient boosting machine (LGBM), extreme gradient boosting (XGB).

**Table 3 cancers-14-03492-t003:** Comparative performance among different sampling strategies in predicting TMB group using LightGBM.

Method	Sensitivity	Specificity	Precision	Accuracy
ADASYN	0.72 ± 0.204	0.7571 ± 0.168	0.7010 ± 0.244	0.7449 ± 0.103
BorderlineSMOTE	0.8 ± 204	0.7143 ± 0.151	0.6573 ± 0.092	0.7462 ± 0.083
RandomOversampler	0.72 ± 0.219	0.8143 ± 0.148	0.7262 ± 0.129	0.7782 ± 0.127
RandomUndersampler	0.8 ± 0.4	0.7393 ± 0.15	0.6810 ± 0.169	0.7641 ± 0.069
SMOTE	0.8 ± 0.32	0.7714 ± 0.19	0.7219 ± 0.258	0.7808 ± 0.11
SVMSMOTE	0.76 ± 0.126	0.8107 ± 0.068	0.7952 ± 0.112	0.7936 ± 0.081

ADASYN: adaptive synthetic, SMOTE: synthetic minority oversampling technique.

**Table 4 cancers-14-03492-t004:** Comparison among different studies on TMB prediction.

Study	Method Summary	Kind of Cancer	Result
Jain et al. [14]	Machine learning algorithm, Image2TMB, integrated three deep learning models.	Lung cancer	auPRC = 0.92 Precision = 0.89
Shi et al. [15]	Deep learning model is based on the ResNet18 architecture.	Lung cancer	AUC = 0.64
Shimada et al. [16]	Convolutional neural network (CNN)-based algorithm.	Colorectal cancer	AUC = 0.934
Tang et al. [23]	LASSO regression selected features. Nomogram model predicted TMB.	Bladder cancer	AUC = 0.853
Liu et al. [24]	Nomogram model predicted TMB.	Lower-grade glioma	AUC = 0.736
The proposed study	The genetic algorithm selected radiomics signatures. LGBM algorithm predicted TMB.	Lower-grade glioma	AUC = 0.7875 auPRC = 0.7556

Only Liu et al. predicted TMB on LGG patients and our proposed study achieved a better performance than this study.

**Table 5 cancers-14-03492-t005:** Hyperparameters of machine learning algorithms in predicting the TMB group.

Algorithm	Optimal Hyperparameters
Logistic Regression	solver = ‘saga’, C = 2.015990003658406, penalty = ‘l1’
Random Forest	‘n_estimators’ = 5, ‘min_samples_split’ = 6, ‘min_samples_leaf’ = 3, ‘max_features’ = ‘auto’, ‘max_depth’ = 30, ‘bootstrap’ = False
Support Vector Machine	kernel = ’rbf’, gamma = 1 × 10^−4^, C = 10
Linear Discriminant Analysis	solver = ‘svd’
Light GBM	learning_rate = 0.005, num_leaves = 15, max_depth = 25, min_data_in_leaf = 15, feature_fraction = 0.6, bagging_fraction = 0.6
XGBoost	max_depth = 1, gamma = 9, colsample_bytree = 0.5, min_child_weight = 1

## Data Availability

We have released our source codes and models at https://github.com/khanhlee/tmb-lgg.

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
