# Peer review of "A Radiomics-Based Machine Learning Model for Prediction of Tumor Mutational Burden in Lower-Grade Gliomas"

_cancers, 2022, doi:10.3390/cancers14143492_

Round 1

Reviewer 1 Report

It has been a very comprehensive study, and I think it has valuable content. The subject is interesting. However, certain aspects related to the technicality and presentation of the paper prevent me from accepting the paper, and I am marking it as "Major revisions requested."

1- Please improve the abstract section; it doesn't deliver enough information about the manuscript to readers.

2- The "Introduction" section needs a major revision to provide a more accurate and informative literature review of the pros and cons of the available approaches and how the proposed method is different comparatively. Also, the motivation and contribution should be stated more clearly.

3- Improve the quality of figures and explain those properly.

4- What makes the proposed method suitable for this unique task? What new development to the proposed method have the authors added (compared to the existing approaches)? These points should be clarified. The discussions should highlight why the proposed method is providing good results. The comparative study from the recently proposed system is missing. The authors compared some old methods. Would you please compare the performance with five new articles?

5- The complexity of the proposed model and the model parameter uncertainty are not mentioned.

6- Explain the novelty of your work presented in this work.

7- Please elaborate more on the proposed approach with more focus on the relations between its components, as they are the core of the solution and need more justification of why to use them.

8- What about the computational complexity (e.g., running time analysis)?

9- Please explain more clearly the limitations and future work.

10- Did the authors use any statistical correction to counteract the problem of multiple comparisons?

11- The logic of the introduction can be improved. For example, the reasons and significance of applying deep learning methods to the neurobiology study of disorders could be introduced. Current progress and critical issues could also be mentioned. The authors can use these articles to edit this section. Predicting tumor mutational burden from histopathological images using multiscale deep learning- A deep learning approach for segmentation of red blood cell images and malaria detection- Computer aided diagnosis system using deep convolutional neural networks for ADHD subtypes-Predicting Tumor Mutational Burden From Lung Adenocarcinoma Histopathological Images Using Deep Learning- Computer Aided Diagnosis System for multiple sclerosis disease based on phase to amplitude coupling in covert visual attention.

Reviewer 2 Report

In 105 lower-grade glioma patients from a public archived database, the authors identified 11 radiomics features, selected by evolution theory inspired LightGBM and genetic algorithm, in a model to classify tumor mutational burden, a biomarker implicated in immunotherapy. Among them, TEXTURE_GLCM_NET_FLAIR_Dis-184 similarity and HISTO_NET_T2_Bin4 were the most productive features in predicting the 185 TBM by SHAP analysis. TEXTURE_GLCM_NET_FLAIR_Dissimilarity differentiated between TMB high and TMB low. The authors also address the data imbalance with six different techniques. Overall this is a well-written manuscript with clear methodology and results. However, novelty is compromised by a paper on LGG patients predicting the 1p/19q co-deletion mutation from MRI features with almost the same machine learning methods, as published in 2021 by the same corresponding author in Cancers (Kha, Quang-Hien, et al. "Development and Validation of an Efficient MRI Radiomics Signature for Improving the Predictive Performance of 1p/19q Co-Deletion in Lower-Grade Gliomas." Cancers 13.21 (2021): 5398.)

Major issues:

Please move descriptions on methods in Results section to Method section instead

What’s the biological/histological interpretation of the two key features identified by SHAP?

Please include limitations of the study in Discussion

Methods: Sharing of source code missing which prevents reproducibility or validation by other groups

Please describe how patients were assigned to either training vs. validation group? By random?

Specific issues:

Lines 65-79: Information in introduction on immunotherapy can be condensed/removed, unless the authors intend to include validation on these immune markers

Line 294: “more intensive display” is more appropriately referred to as “hyperintensity” in radiology terminology

Line 302-304: Blood-brain barrier disruption would lead to contrast enhancement. Please revise sentence

Round 2

Reviewer 1 Report

The manuscript was modified very well. The authors have attempted to address all comments in the revised paper. The manuscript seems acceptable to me for publication in the journal with the corrections made. 

Reviewer 2 Report

The authors have made significant improvement to their manuscript and addressed my concerns satifactorily.